# An Evaluation of Choroidal and Retinal Nerve Fiber Layer Thicknesses Using SD-OCT in Children with Childhood IgA Vasculitis

**DOI:** 10.3390/diagnostics12040901

**Published:** 2022-04-05

**Authors:** Ali Simsek, Mehmet Tekin

**Affiliations:** 1Department of Ophthalmology, School of Medicine, Adıyaman University, Kahta Street, Adiyaman 02000, Turkey; 2Department of Pediatrics, School of Medicine, Adıyaman University, Adiyaman 02000, Turkey; drmehmettekin@hotmail.com

**Keywords:** childhood, choroid, IgA vasculitis, retinal nerve fiber layer

## Abstract

Background: We aimed to evaluate choroidal and retinal nerve fiber layer (RNFL) thicknesses in children undergoing the childhood IgA vasculitis (IgAV). Methods: Fifty-two patients with IgAV aged 1–6 years and 54 healthy children were included. Cases’ age, sex, erythrocyte sedimentation rate (ESR), C-reactive protein (CRP), RNFL thicknesses, and choroidal thickness values were recorded. Results: Median foveal center choroidal thickness was 374.0 µm (315.0 to 452.0 µm) in the IgAV group and 349.5 µm (285.0 to 442.0 µm) in the control group (*p* = 0.001). Median average RNFL thickness was 110.0 µm (91.0 to 134.0 µm) in the IgAV group and 104.0 µm (89.0 to 117.0 µm) in the control group (*p* < 0.001). Choroidal and RNFL thicknesses were significantly greater in all quadrants in the IgAV group than in the control group. No correlation was determined between ESR or CRP and foveal center choroidal and average RNFL thicknesses. Conclusions: Our findings show that choroidal and RNFL thicknesses increased significantly in children undergoing childhood IgA vasculitis compared to the healthy control group. These findings show that the choroid and RNFL are also affected by the inflammatory process in IgAV, which is a systemic vasculitis. We think that the choroidal and RNFL thicknesses can be used as a biomarker for childhood IgAV.

## 1. Introduction

Childhood Immunoglobulin A vasculitis (IgAV), formerly called Henoch–Schönlein purpura, is the most common vasculitis in children. The majority of cases are aged under 10, and there is no gender bias. This systematic, non-granulomatous vasculitis is characterized by the accumulation of immune complexes containing immunoglobulin A in the walls of the small vessels (arterioles, capillary vessels, and venules). Due to its systemic nature, multiorgan involvement is seen in cases of IgAV [1]. Diagnosis of IgAV is based on clinical findings in case of at least one of abdominal pain/gastrointestinal bleeding, arthralgia/arthritis, or hematuria/proteinuria accompanying palpable purpura, particularly in the lower extremities (Figure 1) [2].

Uveitis is one of the main clinical findings in Behçet’s disease, another systemic vasculitis, and choroidal thickening has been reported in patients with uveitis [3,4]. However, choroidal thinning has been reported in non-uveitis Behçet’s disease, and subclinical choroidal involvement must not be overlooked [5]. Increased subfoveal choroidal thickness has also been reported in patients with polyarteritis nodosa, another systemic vasculitis [6].

The choroid is a densely vascularized structure consisting of a network of capillaries and larger vessels with several functions, including supplying oxygen and other nutrients to the outer layers of the retina and retinal pigment epithelium, removing metabolic waste products from the outer retina, and temperature regulation [7]. The layers of the retina can be examined in detail using spectral domain optical coherence tomography (SD-OCT), and the choroid, a vascular bed nourishing the retina, can be evaluated using enhanced depth imaging optical coherence tomography (EDI-OCT) [8]. Currently, SD-OCT is the gold standard non-invasive imaging technique for the diagnosis of clinical ocular diseases such as age-related macular degeneration, diabetic retinopathy, central serous chorioretinopathy, and inherited retinal diseases. Visualization of the choroid, which plays a role in the pathogenesis of retinal diseases, was possible with the advent of SD-OCT. In addition, it is possible to measure the physical and optical parameters of the eye, including the axial length and lens thickness, with the optical biometer, which is a variant of SD-OCT [9].

In recent years, measurement of choroidal and retinal nerve fiber layer (RNFL) thick-nesses with SD-OCT has been suggested as an inflammatory marker for different systemic diseases with vascular involvement, such as lupus, familial Mediterranean fever, and Behçet’s disease [10]. Despite being a systemic vasculitis, to the best of our knowledge, no previous studies have examined choroidal and RNFL thicknesses in patients with IgAV. The aim of this study is to determine whether choroidal and RNFL thicknesses change in childhood IgAV and to determine whether these thicknesses can be used as an inflammatory marker in IgAV.

## 2. Materials and Methods

### 2.1. Study Design

This prospective cross-sectional study was performed between 1 September 2015 and 31 August 2019 at the Adıyaman University School of Medicine ophthalmology and pediatric clinics, Turkey. Approval for the study was granted by the Adıyaman University Medical Faculty Non-Interventional Ethical Committee (No. 2015/05-15). Procedures at all stages of the study were carried out in compliance with the principles of the Declaration of Helsinki. Informed consent forms were received from all participants or their parents before the study commenced.

Fifty-two patients with IgAV aged one to six years and fifty-four healthy children were included in the study. Diagnosis of IgAV was based on at least one of abdominal pain/gastrointestinal bleeding, arthralgia/arthritis, or hematuria/proteinuria findings accompanying palpable purpura in the lower extremities. Healthy children selected from among individuals similar to the IgAV group in terms of age and sex and presenting for routine health checks were enrolled in the control group. Patients with accompanying thrombocytopenia, anemia, malaria, syphilis, bleeding or clotting disorders, acute kidney failure (reduced glomerular filtration rate (GFR): <80 mL/min/1.73 m^2^), massive proteinuria (>40 mg/m^2^/day) or sepsis/disseminated intravascular coagulation, medical histories of systemic lupus erythematosus, polyarteritis nodosa, coagulation disorder, diabetes mellitus, renal disease, smoking, drinking, immunosuppressive/cytotoxic drug use, contact lens use, ocular trauma, keratoconus, glaucoma, topical drug use, or eye surgery, or with amblyopia, strabismus, myopia, hypermetropia, or with an astigmatic refractive error exceeding ±1.0 D at ocular examination were excluded from the study. Choroidal thickness is influenced by a number of factors including age, gender, axial length, retinal thickness, and intraocular pressure (IOP) [11,12]. Similarly, RNFL thickness is influenced by age, axial length, and spherical equivalent [13]. Therefore, the patient and the control groups were selected from cases matched in terms of age, gender, refractive error, and axial length.

#### Ophthalmological Examinations

Eye examinations were performed in the first 48 h following diagnosis of IgAV. Ophthalmological examination including IOP, refractive error, split-lamp biomicroscopy, and visual acuity measurement and dilated fundus examination were performed in all cases. Best visual acuity values (as measured on a Snellen chart, decimal fraction) were recorded. Central corneal thickness (CCT) and axial length were measured using an optical biometer (Lenstar LS 900; Haag Streit AG, Koeniz, Switzerland). The RNFL and the choroid were visualized in non-dilated pupils using SD-OCT (Spectralis; Heidelberg Engineering, Heidelberg, Germany). The scan quality ranged from no signal (0) to excellent (40), and only high-quality images (a well-focused optic disc with a signal strength >20 Db and centered) were selected. The variation in the accuracy of ranging (length measurement) expected from using a different OCT machine is typically caused by the changes in the calibration and image reconstruction algorithm adopted for various OCT/biometer machines [14]. All children were examined for choroidal thickness at the same time period between 8:00 a.m. and 10:00 a.m. to avoid diurnal variations [15]. Right eye values were used for statistical analyses. To improve the visualization of the choroid, the instrument’s enhanced depth imaging mode was used in combination with automatic real-time eye tracking and frame averaging. The wavelength of the OCT instrument was 870 nm, with an axial resolution of 5 μm and a scanning speed of 40,000 A-scans per second. Each radial OCT image was the average of 30 B-scans. [16]. Each subject was imaged by both ophthalmologists (A.S. and A.A.Y.) who were blinded to the clinical information of the examined eyes, and the two values, captured by two different ophthalmologists, were averaged for analysis. Inter-examiner reproducibility of all manual measurements was evaluated by the intraclass correlation coefficient (ICC) so that values of greater than 0.80 were accepted as good agreement. Choroidal thickness was defined as the distance between the hyper reflective line and the outer retinal pigment epithelium line behind the large-sized choroidal vessel layers at the scleral interface. We manually measured the thicknesses at seven different points: at the foveal center and within the horizontal, temporal, and nasal quadrants at 500-μm intervals as far as 1500 μm from the foveal center. RNFL thickness was defined from the optic nerve head scan. A volumetric scanning protocol was used, imaging a 15 by 15 region surrounding the optic nerve head (circle scan size 3.4 mm). The average RNFL thickness and those in the four quadrants were automatically calculated by the SD-OCT (superior, nasal, inferior, and temporal, 90° each).

### 2.2. Data Acquisition

Cases’ age, sex, arterial blood pressure, complete blood count, erythrocyte sedimentation rate (ESR), C-reactive protein (CRP), creatinine, albumin, complete urine examination, urinary protein level, GFR, occult blood in stool test, IOP, visual acuity, CCT, axial length, and choroidal and RNFL thicknesses were recorded.

### 2.3. Statistical Analysis

Statistical Package for the Social Sciences 21.0 software (SPSS Inc., Chicago, IL, USA) was used for statistical analyses. The chi-square test was used to compare categorical variables. The Kolmogorov–Smirnov test was performed to determine whether the continuous data were normally distributed. The independent Student t test was used for comparison of normally distributed data, and these results were expressed as mean ± standard deviation. Non-normally distributed data were compared with the Mann–Whitney U test and values were expressed as median (minimum to maximum). Pearson correlation analysis was applied to determine a linear relationship between CRP and ESR and choroidal and RNFL thicknesses. *p* values < 0.05 were considered statistically significant.

## 3. Results

Fifty-eight patients with IgAV were originally included in the study, but two were excluded due to nephrotic proteinuria secondary to IgAV, one because of development of acute kidney failure (eGFR: <80 mL/min/1.73 m^2^), two due to pronounced refractive error, and one because a high-quality image could not be obtained. The study thus continued with 52 patients with IgAV and 54 healthy children. Girls constituted 24 (46.2%) of the IgAV group and boys 28 (53.8%), while 25 (46.3%) of the control group were girls and 29 (53.7%) were boys. Mean ages were 7.1 ± 2.4 years in the IgAV group and 7.8 ± 2.3 in the control group. There was no significant difference between the groups in terms of sex or age (*p* = 0.571 and *p* = 0.154, respectively). Median CRP was 3.20 (0.01–15.30) mg/dL and median ESR 16 (5–32) mm/h in the IgAV group, compared to 0.40 (0.01–3.50) mg/dL and 7 (3–15) mm/h in the control group. CRP and ESR values were significantly higher in the IgAV group (*p* < 0.001 for both). No significant difference was determined between the groups in terms of hemoglobin, creatinine, or albumin values. No difference was also determined between the groups in terms of IOP, visual acuity, axial length, or CCT (Table 1). No cotton-wool spot, conjunctivitis, uveitis, keratitis, scleritis, episcleritis, optical neuritis, macular edema, papilledema, or retinal hemorrhage/infarct was observed in either group.

Foveal center choroidal thicknesses measured using EDI-OCT were 374.0 µm (315.0 to 452.0 µm) in the IgAV group (Figure 2) and 349.5 µm (285.0 to 442.0 µm) in the control group (*p* = 0.001). Choroidal thicknesses in all quadrants in the IgAV group were significantly greater than in the control group (Table 2).

Median average RNFL thicknesses were 110.0 µm (92.0 to 134.0 µm) in the IgAV group (Figure 3) compared to 104.0 µm (89.0 to 117.0 µm) in the control group (*p* < 0.001). RNFL thicknesses were significantly higher in all quadrants in the IgAV group compared to the control group (Table 3).

Pearson correlation analysis revealed no correlation between ESR and CRP and foveal center choroidal or average RNFL thicknesses (Table 4).

## 4. Discussion

The results of this study showed a significant increase in choroidal and RNFL thicknesses in patients with IgAV compared to the healthy control group of similar age and gender. Choroidal and RNFL thicknesses increased in all quadrants in the IgAV patients.

IgAV is the most common form of vasculitis in childhood. The disease causes multisystem involvement, particularly urinary, digestive, cutaneous, and locomotor. In addition to classic findings such as abdominal pain/gastrointestinal bleeding, arthralgia/arthritis, and hemorrhage/proteinuria accompanying palpable purpura in the lower extremities, the disease can also cause pathologies such as intussusception, cerebral hemorrhage, seizure, myocarditis, myositis, and pulmonary hemorrhage [1].

In IgAV, IgA immune complexes are transported through the blood and adhere to small vessel walls, then cause an oxidative burst of neutrophils and secreted enzymes, and oxygen radicals cause disruption of the vessel basement membrane, resulting in endothelial damage [17]. The choroidal vasculature is the primary source of both oxygen and nutrients to the outer retina, including the retinal pigment epithelium, photoreceptors, and, otherwise, avascular fovea [7]. This high flow system also provides primary ocular access for the circulating immune complexes, therefore, choroid is a common site of inflammation [18]. It can be hypothesized that the choroid and RNFL, neural tissue of the eye fed by the choroid, may be influenced due to circulating IgA immune complexes. Based on these observations, we examined the choroidal and RNFL thicknesses with SD-OCT in IgAV patients.

For many years, the choroid could only be evaluated by indocyanine green angi-ography, ultrasound, and laser Doppler flowmetry. Although these techniques are useful for determining vessel abnormalities or changes in the choroidal blood flow, it was not possible to visualize three-dimensional anatomical information about the choroid [19]. More recently, technological improvements in structural OCT have revealed important morphological and functional characteristics of the choroid in physiological and pathological conditions, including inflammatory disease. The enhanced depth imaging OCT is a noninvasive technique that provides high-resolution cross-sectional three-dimensional images of the retina and RNFL and good repeatability of choroidal thickness measurements [20].

Uveitis, keratitis, episcleritis, and optical neuritis can be seen in the vasculitic condition Behçet’s disease [21,22]. Cotton-wool spots, anterior ischemic optic neuropathy, central retinal artery occlusion, posterior scleritis, and choroiditis have been reported in patients with polyarteritis nodosa [23]. Bilateral conjunctivitis, uveitis, iridocyclitis, superficial punctate keratitis, vitreous opacities, and papilledema can be seen in Kawasaki disease [24]. Optical neuritis, retinopathy, and choroidopathy have been reported in systemic lupus erythematosus [23,25]. There have also been case reports of bilateral cystoid macular edema, cotton-wool spots, and anterior ischemic optic neuropathy in patients with IgAV [26]. No cotton-wool spot, conjunctivitis, uveitis, keratitis, scleritis, episcleritis, optical neuritis, macular edema, papilledema, or retinal hemorrhage/infarct were determined in the IgAV patients in the present study.

Chung et al. [27] reported increased choroidal thicknesses and suggested that choroidal thickness can be used as an indicator of subclinical ocular inflammation and systemic inflammation in Behçet’s disease. Tetikoğlu et al. [28] reported increased choroidal thicknesses in patients with rheumatoid arthritis. Baytaroğlu et al. [6] reported increased choroidal thicknesses in patients with polyarteritis nodosa and that choroidal thickness can be employed as an indicator of subclinical systemic involvement. Some authors focused on choroidal thickness as a biomarker for cardiovascular disease risk factors [29]. Arnold et al. [30] reported that longer axial length and older age are associated with thinner subfoveal choroidal thickness, but choroidal thickness was not significantly associated with sex or cardiovascular history. In the present study, choroidal thicknesses were significantly higher in all quadrants in the patients with IgAV compared to the healthy control group. The patient and the control groups were selected from cases matched in terms of age, gender, refractive error, and axial length.

Liu et al. [31] reported significant RNFL thinning in systemic lupus erythematosus, and they speculated that OCT measurements may be indicative of neurodegeneration in systemic lupus erythematosus. Tetikoğlu et al. [28] reported no significant difference be-tween the rheumatoid arthritis and healthy group regarding RNFL. Bayram et al. [32] re-ported that RNFL thickness increased significantly in patients with COVID-19 disease compared to the healthy group. Arnould et al. [33] reported that decreased retinal vessels’ calibers were associated with a decreased RNFL thickness in the elderly without optic neuropathy. In the present study, RNFL thicknesses were significantly higher in all quad-rants in the patients with IgAV compared to the healthy control group. We hypothesized that the increased RNFL thickness may be linked to IgA immune complexes transported through the choroid.

Additionally, no correlation was found between CRP and ESR and choroidal or RNFL thicknesses in the present study.

One of the principal limitations of the present study, which is, to the best of our knowledge, the first to investigate choroidal and RNFL thicknesses in patients with IgAV, is the relatively low number of cases. In addition, it is unknown whether the increase in choroidal and RNFL thicknesses will cause clinically detectable ocular symptoms in later years.

In conclusion, our findings show that choroidal and RNFL thicknesses increased significantly in children undergoing IgAV compared to the healthy control group. These findings show that the choroid and RNFL are also affected by the inflammatory process in IgAV, which is a systemic vasculitis. We think that the choroidal and RNFL thicknesses can be used as a biomarker for childhood IgAV. It is known that IgAV can cause kidney damage in the long term, but there are no data on its effects on vision. Our findings now need to be confirmed with larger patient groups, and the long-term clinical effects of subclinical choroidal and RNFL thickness increases also need to be observed.

## Figures and Tables

**Figure 1 diagnostics-12-00901-f001:**
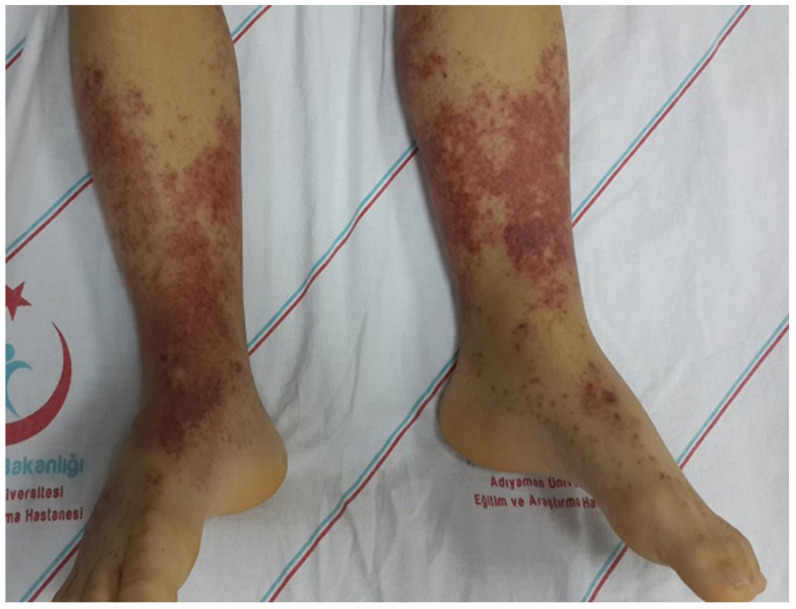
A child with palpable purpura in the lower extremities.

**Figure 2 diagnostics-12-00901-f002:**
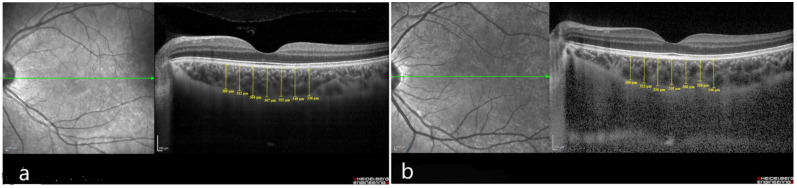
(**a**) Choroidal thickness measurements obtained by spectral-domain optical coherence tomography in a child with immunoglobulin A vasculitis. (**b**) Choroidal thickness measurements in a child from the control group. Images were obtained by averaging multiple B-scans from same position in both groups.

**Figure 3 diagnostics-12-00901-f003:**
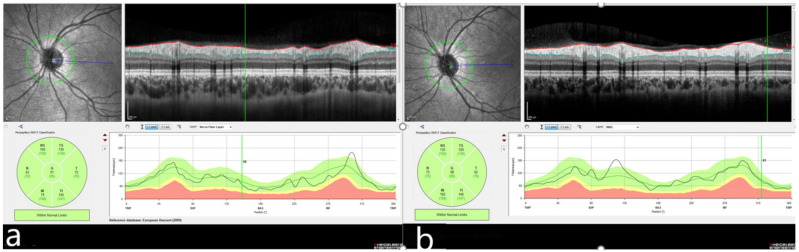
(**a**) Retinal nerve fiber layer thickness measurements obtained by spectral-domain optical coherence tomography in a child with immunoglobulin A vasculitis. (**b**) Retinal nerve fiber layer thickness in a child from the control group. Images were obtained by averaging multiple B-scans from the same position in both groups.

**Table 1 diagnostics-12-00901-t001:** Demographic, laboratory, and ocular characteristics of the groups.

	IgAV Group (n = 52)	Control Group (n = 54)	*p*
Gender (F/M) ^#^	24/28	25/29	0.571
Age (year) ^†^	7.1 ± 2.4	7.8 ± 2.3	0.154
Hemoglobin (g/dL) ^†^	11.9 ± 0.9	11.8 ± 0.8	0.676
CRP (mg/dL) ^¶^	3.20 (0.01–15.30)	0.40 (0.01–3.50)	**<0.001 ***
ESR (mm/h) ^¶^	16 (5–32)	7 (3–15)	**<0.001 ***
Creatinine (mg/dL) ^†^	0.60 ± 0.18	0.57 ± 0.14	0.390
Albumin (mg/dL) ^†^	3.54 ± 0.30	3.48 ± 0.28	0.334
IOP (mmHg) ^†^	14.2 ± 1.3	13.7 ± 1.2	0.054
Visual acuity (Snellen chart, in decimal) ^†^	1.14 ± 0.12	1.17 ± 0.09	0.147
CCT (μm) ^†^	532.4 ± 20.4	530.7 ± 26.0	0.704
Axial length (mm) ^†^	22.02 ± 0.78	22.06 ± 0.83	0.817

* *p* < 0.05; ^#^ Chi-square test; ^†^ Independent Student *t* test, mean ± standard deviation; ^¶^ Mann–Whitney U test, median (minimum–maximum); IgAV, immunoglobulin A vasculitis; CRP, C-reactive protein; ESR, erythrocyte sedimentation rate; IOP, intraocular pressure; CCT, central corneal thickness.

**Table 2 diagnostics-12-00901-t002:** Comparisons of Choroidal Thicknesses in the Right Eyes of Immunoglobulin A Vasculitis and Control Groups.

	IgAV Group (n = 52)	Control Group (n = 54)	*p*
Foveal center, µm	374.0 (315.0–452.0)	349.5 (285.0–442.0)	**0.001 ***
Temporal, 1500 µm	320.5 (290.0–400.0)	300.0 (251.0–400.0)	**0.016 ***
Nasal, 1500 µm	302.5 (233.0–433.0)	290.0 (220.0–370.0)	**0.003 ***
Temporal, 1000 µm	342.5 (234.0–420.0)	319.0 (254.0–420.0)	**0.009 ***
Nasal, 1000 µm	324.0 (261.0–418.0)	300.0 (231.0–398.0)	**0.002 ***
Temporal, 500 µm	359.0 (301.0–437.0)	332.5 (258.0–437.0)	**0.004 ***
Nasal, 500 µm	345.0 (250.0–493.0)	316.0 (245.0–410.0)	**0.003 ***

* *p* < 0.05 Mann–Whitney U test was used for comparisons and results were expressed as median (minimum to maximum) values; IgAV, immunoglobulin A vasculitis.

**Table 3 diagnostics-12-00901-t003:** Comparisons of Retinal Nerve Fiber Layer Thicknesses in the Right Eyes of Immunoglobulin A Vasculitis and control groups.

	IgAV Group (n = 52)	Control Group (n = 54)	*p*
Nasal inferior RNFL, µm	124.0 (115.0–170.0)	113.0 (97.0–127.0)	**<0.001 ***
Nasal superior RNFL, µm	116.0 (108.0–153.0)	106.0 (94.0–122.0)	**<0.001 ***
Nasal RNFL, µm	86.0 (67.0–116.0)	75.0 (67.0–98.0)	**<0.001 ***
Temporal inferior RNFL, µm	162.5 (135.0–189.0)	147.0 (129.0–175.0)	**<0.001 ***
Temporal superior RNFL, µm	149.5 (123.0–179.0)	137.0 (115.0–165.0)	**0.001 ***
Temporal RNFL, µm	89.0 (70.0–117.0)	82.5 (59.0–107.0)	**0.001 ***
Average RNFL, µm	110.0 (92.0–134.0)	104.0 (89.0–117.0)	**<0.001 ***

* *p* < 0.05 Mann–Whitney U test was used for comparisons and were expressed as median (minimum to maximum) values; IgAV, immunoglobulin A vasculitis; RNFL, retinal nerve fiber layer.

**Table 4 diagnostics-12-00901-t004:** Correlation between inflammation markers and choroidal and retinal fiber layer thicknesses.

	*r*	*R^2^*	Estimated Equation	95% CI	*p*
ESR-Foveal center CT	−0.12	1.551*E*−4	*y* = 3.75*E*2 − 0.06 * *x*	−0.213 to +0.198	0.930
ESR-Average RNFL	0.126	0.016	*y* = 1.08*E*2 + 0.18 * *x*	−0.203 to +0.425	0.372
CRP-Foveal center CT	−0.013	1.773*E*−4	*y* = 3.75*E*2 − 0.12 * *x*	−0.282 to +0.232	0.925
CRP-Average RNFL	−0.260	0.068	*y* = 1.14*E*2 − 0.67 * *x*	−0.463 to +0.013	0.062

Pearson correlation was used for comparison; 95% CI, confidence interval; ESR, erythrocyte sedimentation rate; CRP, C-reactive protein; CT, choroidal thickness; RNFL, retinal nerve fiber layer.

## Data Availability

The data that support the findings of this study are available from the corresponding author, upon reasonable request.

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
