# Peer review of "An Evaluation of Choroidal and Retinal Nerve Fiber Layer Thicknesses Using SD-OCT in Children with Childhood IgA Vasculitis"

_diagnostics, 2022, doi:10.3390/diagnostics12040901_

Round 1

Reviewer 1 Report

Behçet’s disease and HSP belong to the category of systemic vasculitis. Behcet's disease can accumulate large, medium and small blood vessels, and HSP still belongs to small vasculitis. This paper studies HSP, so there is no need to discuss too much for Behçet’s disease in the discussion part. In addition, because smoking and drinking and diurnal variations in choroidal structure can affect choroidal thickness, whether smoking and drinking were excluded from the inclusion criteria, and whether all children were examined for choroidal thickness at the same time period?

Author Response

Reviewer 1:

Behçet’s disease and HSP belong to the category of systemic vasculitis. Behcet's disease can accumulate large, medium and small blood vessels, and HSP still belongs to small vasculitis. This paper studies HSP, so there is no need to discuss too much for Behçet’s disease in the discussion part. In addition, because smoking and drinking and diurnal variations in choroidal structure can affect choroidal thickness, whether smoking and drinking were excluded from the inclusion criteria, and whether all children were examined for choroidal thickness at the same time period?

Response: Since there are few OCT studies on vasculitides in the literature and these studies are generally related to Behçet's disease, studies on Behçet's disease were slightly over-referenced. In line with your suggestion, the discuss about Behçet's disease has been reduced in the discussion section. Smoking and drinking were added to the exclusion criteria. It was added to the method section that all children were examined for choroidal thickness at the same time period.

Reviewer 2 Report

In this manuscript, authors have evaluated the choroidal and retinal nerve fiber layer (RNFL) thicknesses in children undergoing the systemic vasculitis Henoch–Schönlein purpura (HSP). Henoch–Schönlein purpura (HSP) is the most common vasculitis in children mostly reported  under age 10 without any gender bias. They performed clinical study on 54 healthy and 52 children with HSP. This is a reliable number for statistically reliable results and findings. Authors conclude that  choroidal and RNFL thicknesses increased significantly in children with HSP relative to healthy children. This is an interesting piece of clinical work and the manuscript is written generally well. Authors need to address the following concerns prior to its full acceptance.

  1. Authors may provide a general statement about the significance of OCT as an optical non-invasive imaging technique which is referred to as the gold standard for the clinical diagnostics of ocular diseases . Optical biometer is a variant of OCT which is dedicated for measuring the physical and optical parameters of eye including axial length, lens thickness etc. This will provide a more general idea of these technologies covering wide range of readers. (doi: 10.1097/ICU.0b013e32835f8bf8)
  2. There is a huge difference in image quality between Figure 2 (a) and (b). Did the Fig 2 (b) is obtained by averaging multiple B-scan s from same position? If yes, please mention that and indicate number of frames used for averaging ? I always recommend to use averaged B-scans since it allows significant enhancement of the image quality by supressing inherent speckle noise in OCT images. Moreover, this would allow users to segment retina layers and choroid more precisely using dedicated retinal layer segmentation programs. Authors are advised use averaged Bscan for Fig 2(a)  as well. Apply it for 3 (a) and 3(b) as well.
  3. Authors are advised to mention about the variation of the accuracy of ranging (length measurement) expected from using a different OCT machine typically caused by the changes in the calibration and image reconstruction algorithm adopted for various OCT/biometer machines. (https://doi.org/10.1088/0031-9155/61/21/7652, https://doi.org/10.1088/1612-2011/12/5/055601) This would justify the discrepancy in the measurements repeated using different machines. 

4. I can understand from Figure 3 that authors have used automatic segmentation program for measuring RFNL thickness measurements. Could you justify why don't you use the same program for choroidal thickness measurements instead of using line-based thickness measurements. I was wondering since mist of commercially available systems have this feature with image analysis package. 

Author Response

Reiewer 2:

In this manuscript, authors have evaluated the choroidal and retinal nerve fiber layer (RNFL) thicknesses in children undergoing the systemic vasculitis Henoch–Schönlein purpura (HSP). Henoch–Schönlein purpura (HSP) is the most common vasculitis in children mostly reported  under age 10 without any gender bias. They performed clinical study on 54 healthy and 52 children with HSP. This is a reliable number for statistically reliable results and findings. Authors conclude that  choroidal and RNFL thicknesses increased significantly in children with HSP relative to healthy children. This is an interesting piece of clinical work and the manuscript is written generally well. Authors need to address the following concerns prior to its full acceptance.

  1. Authors may provide a general statement about the significance of OCT as an optical non-invasive imaging technique which is referred to as the gold standard for the clinical diagnostics of ocular diseases . Optical biometer is a variant of OCT which is dedicated for measuring the physical and optical parameters of eye including axial length, lens thickness etc. This will provide a more general idea of these technologies covering wide range of readers. (doi: 10.1097/ICU.0b013e32835f8bf8)

Response: General information about SD-OCT technology has been added to the text in line with your suggestion.

  1. There is a huge difference in image quality between Figure 2 (a) and (b). Did the Fig 2 (b) is obtained by averaging multiple B-scan s from same position? If yes, please mention that and indicate number of frames used for averaging ? I always recommend to use averaged B-scans since it allows significant enhancement of the image quality by supressing inherent speckle noise in OCT images. Moreover, this would allow users to segment retina layers and choroid more precisely using dedicated retinal layer segmentation programs. Authors are advised use averaged Bscan for Fig 2(a) as well. Apply it for 3 (a) and 3(b) as well.

Response: It was clearly stated that the images in both groups obtained by averaging multiple B-scan s from same position. Avaraged B-scans images were also used for Figures 2a, 3a and 3b.

  1. Authors are advised to mention about the variation of the accuracy of ranging (length measurement) expected from using a different OCT machine typically caused by the changes in the calibration and image reconstruction algorithm adopted for various OCT/biometer machines. (https://doi.org/10.1088/0031-9155/61/21/7652, https://doi.org/10.1088/1612-2011/12/5/055601) This would justify the discrepancy in the measurements repeated using different machines.

Response: We were mentioned about the variation of the accuracy of ranging (length measurement) expected from using a different OCT machine typically caused by the changes in the calibration and image reconstruction algorithm adopted for various OCT/biometer machines.

  1. I can understand from Figure 3 that authors have used automatic segmentation program for measuring RFNL thickness measurements. Could you justify why don't you use the same program for choroidal thickness measurements instead of using line-based thickness measurements. I was wondering since mist of commercially available systems have this feature with image analysis package.

Response: As of the date we started the study (2015), there was no software in our device for line-based thickness measurements for choroidal thickness measurements. Therefore, we could not use line-based thickness measurements in the choroid.